# Skatescape in the Making: Developing Sustainable Urban Pedagogies through Transdisciplinary Education

**Kirsi Pauliina Kallio** [1,*], **Salla Jokela** [2], **Mikko Kyrönviita** [3], **Markus Laine** [3] **and Jonathon Taylor** [4]

1   Faculty of Education and Culture, Tampere University, 33014 Tampere, Finland
2   Faculty of Social Sciences, Tampere University, 33014 Tampere, Finland; salla.e.jokela@tuni.fi
3   Faculty of Management and Business, Tampere University, 33014 Tampere, Finland; mikko.kyronviita@tuni.fi (M.K.); markus.laine@tuni.fi (M.L.)
4   Faculty of Built Environment, Tampere University, 33720 Tampere, Finland; jonathon.taylor@tuni.fi
*   Correspondence: kirsipauliina.kallio@tuni.fi; Tel.: +358-40-190-4025

**Abstract:** The current trend of higher education for sustainable urban development links with parallel developments in urban governance and environmental pedagogy. Many programs and policies identify cities and citizens as key drivers of change for sustainable futures, however scholarly work on the related pedagogies is lacking. These endeavors are clearly present in the Tampere city-region, Finland. Supported by national educational and city-regional strategies, the city is promoting sustainable lifestyles and infrastructures by means of multistakeholder governance including citizen participation and sustainability education. This paper analyzes, as a case study of transdisciplinary sustainability education, a collaboration between Tampere University, a skateboarding high school, and the Hiedanranta urban district developed as a real-life laboratory of sustainable urban development. We explore the pedagogical dimensions of the collaboration by drawing from the theoretical perspective of 'positive recognition' and conceiving the *Hiedanranta skatescape*—a socio-physical entity formed around skateboarding—as a 'boundary object'. The paper suggests urban environments act as boundary objects that enable productive collaboration between various actors when informed by pedagogies of positive recognition. In conclusion, we propose that the value of this approach, binding together multistakeholder governance and transdisciplinary learning, lies in its capacity to encourage novel forms of sustainable agency.

**Keywords:** sustainable urban development; sustainable higher education; sustainable high school; sustainable pedagogy; education for sustainable development; multistakeholder governance; skateboarding; citizen participation; positive recognition; boundary object





## 1. Introduction

Education for sustainable urban development is an emerging global trend in higher education [1–5], along with parallel developments in urban governance [6,7] and environmental pedagogy at all educational levels [8,9]. A common aim in these initiatives is to move towards a more sustainable world by transforming cities and activating urban citizens as key drivers of change. The United Nations Sustainable Development Goals (SDGs) [10] sets a framework for contextualized initiatives, within the dimensions of environmental, social, economic, and cultural sustainability.

These endeavors are clearly present in the Tampere city-region, the second-largest urban agglomeration in Finland. The city's *Sustainable Tampere 2030* program includes ambitious goals, for instance carbon neutrality, to be implemented in broad multistakeholder collaboration with the city's service areas, various departments, public utilities, companies, and citizens [11]. Citizen engagement includes creative and ludic approaches, such as a mobile game that encourages players to interact and move towards a climate friendly lifestyle [12]. Other forms of co-creation for sustainability include collaborative planning, participatory budgeting, hackathons, and various 'agile experiments' with businesses and

residents to develop new solutions in existing and emerging urban environments. The city's enabling role in urban development also supports grassroots organizations and small-scale initiatives, with the aim of diversifying participation opportunities and engaging citizens who find formal participatory roles uninviting or inaccessible—the young, for instance. One experimental arena, the new urban district of Hiedanranta, is being developed as a real-life laboratory of sustainable urban development.

At the same time, institutions of education in Tampere are changing their programs and practices to contribute to sustainability aims. These local activities are strongly supported by national policies, most notably school curriculums and higher education strategies where sustainability, in connection with other environmental themes, is emphasized in the latest reforms. The co-operational organization for Finnish universities, *Universities Finland* [13], recently launched a common agenda of sustainable development in the form of 12 theses that cover the key measures through which universities will respond to the SDGs. Among other things, the accessibility of higher education is to be improved, and universities should " . . . share sustainable and responsible practices openly and actively and develop them together [as] bold, committed and responsible partners".

The HE agenda that encourages collaboration between education institutions and local governance is linked with the renewed high school curriculum, effective from autumn 2021 and preceded by secondary, primary, and early education reforms that also clearly set sustainability aims in accordance with the pupils' maturity. According to the high school curriculum reform, "The student learns the basics of ecologically, economically, socially and culturally sustainable lifestyles and the interdependences thereof. The student understands why human activity must be adjusted to the carrying capacity of natural environments, to limited natural resources and their sustainable use. Experiences of care regarding people and the nature strengthen trust in the effectiveness of mundane good deeds. The student familiarizes with scientific knowledge and practices related to climate change mitigation and preserving biodiversity, and is offered opportunities to observe, plan, study and assess the activities through which these phenomena can be transformed toward sustainability. The student reflects upon the observations with regard societal impact, to identify structures that enable or hinder the sustainable activities of different communities." ([14], translation KPK).

To achieve these goals, and in accordance with internationally defined key competencies in sustainability, transdisciplinary practices are embraced at different levels of education. By engaging in ongoing urban projects and related problem solving, students learn to cross disciplinary boundaries, which enhances their competences in the interplay between natural scientific facts, social scientific understanding, and practical societal knowledge [5,15,16]. In this kind of cross-disciplinary learning the idea of approaching the same problem or phenomenon from different perspectives holds center stage, which includes the recognition of various values, worldviews, societal positions, and ways of doing things. The goal is to create understanding of—and solutions to—multifaceted challenges, even when an all-encompassing consensus about the nature of those challenges cannot be formed.

These parallel developments in higher education, urban governance, and environmental education form a strong basis for sustainable urban development that combines novel participatory pedagogical and governance perspectives. However, to date, few studies have described how transdisciplinary teaching of sustainability in higher education can be achieved in practice [17,18]. To contribute to this very development, we introduce a case of transdisciplinary university 'urban lab' course, developed through collaboration of Sustainable Urban Development and Environmental Pedagogy programs at Tampere University; Sampo Upper Secondary School; the Hiedanranta Development Programme responsible for developing a new city district in Tampere; and a local skaters' association, Pirkanmaan Kaarikoirat. While the implementation is yet to begin, the key collaborative structures and the leading pedagogical ideas informing the project largely exist. The focus of the paper hence lies in these *framings* of the sustainable urban pedagogy project, which will generate

further research upon its realization. Specifically, we ask what transdisciplinary teaching of sustainability in higher education can achieve when informed by pedagogies of positive recognition and the conceptual framing of the boundary object.

The paper first describes the urban development framework based on multistakeholder governance and the related development of skating culture and infrastructures in the Tampere city-region. The context of our pedagogical project, *Hiedanranta skatescape,* is framed after that. This is followed by the presentation of our transdisciplinary collaboration and the co-creative research approach, followed by a brief introduction to the pedagogies of 'positive recognition'. Finally, we discuss our case as an example of *pedagogically useful boundary object in sustainable higher education*, where mundane urban politics and multi-stakeholder urban governance are brought together in a productive manner, in the form of co-creative transdisciplinary and cross-institutional education. The paper ends with brief conclusions summarizing the key arguments.

## 2. Developing Sustainability through Skateboarding Culture

### 2.1. Hiedanranta as a Sustainability Initiative in the Tampere City-Region

Hiedanranta, located four kilometers from the Tampere City Centre (Figure 1), is an old industrial area, and a planned location of a new district of 25,000 residents and 10,000 working places. The Development Programme of Hiedanranta is led by *Hiedanrannan Kehitys Oy,* a company owned by the City of Tampere, in collaboration with the City of Tampere's Planning Department, landowners, and other stakeholders. The aim is to transform the former pulp mill and the surrounding industrial area into a diverse city district that combines housing, work, leisure, and services. The plans describe a focus on circular economy, resource-efficiency, and smart and sustainable solutions to mobility, buildings, services, and infrastructure enabling "a sustainable lifestyle" for future residents [19]. The area will be traversed by the Tampere Tram, the flagship of the sustainable city-regional traffic plan, currently under construction and planned to be extended to Hiedanranta by 2024 [20].

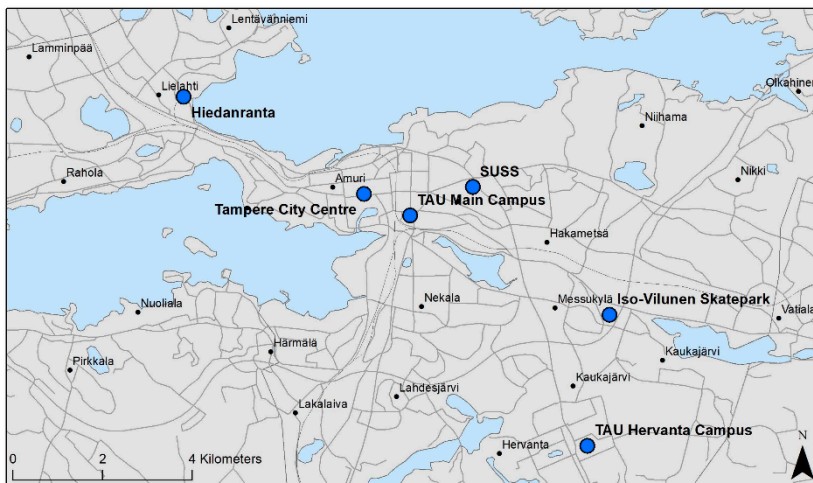

**Figure 1.** Location of the Hiedanranta district in relation to the city center and other districts, Tampere University (TAU) campuses, Sampo Upper Secondary School (SUSS), and Iso-Vilunen skatepark in Tampere, Finland.

The Hiedanranta Development Programme involves all key elements included in the national strategy of urban development, stressing green transition in urban planning, housing, transportation, and the organization of services, implemented through 'MAL agreements' in the major city-regions. The current Finnish Government's Programme [21] states, "We will continue the urban development partnerships between the central government and large urban regions through 'Land use, housing and transport' (MAL) agreements and make the agreements more binding. We will build carbon neutral urban regions, boost

the volume of housing production and increase the proportion of sustainable means of transport". Against this context, the sustainable development of Hiedanranta reflects current national government policy, with a local vibe.

The Development Programme is a flat organization with its own budget, which has enabled agile ways to pilot new ideas—for example, programs aimed at empowering young people, reducing inequality and segregation—while mitigating and tackling the negative impacts of climate change in alignment with the city's goal of carbon neutrality by 2030 [22]. This is a highly ambitious yet achievable goal, even if the roadmap to carbon neutral Tampere 2020 reveals that 20% of the reduction means are still unidentified [23]. Hiedanranta has an important role as a real-world urban laboratory and testbed for seeking solutions to challenges of sustainable urban development, including both environmental challenges (e.g., biodiversity depletion and management of rubble from construction sites) and social challenges (e.g., income inequality, segregation, and marginalization of young people). Successful pilots can be later applied city-regionally.

Through the city's enabling approach, the Hiedanranta area and the empty industrial buildings have been opened for citizens, artists, handicraft entrepreneurs, event organizers, grassroots organizations, and companies, to test new ideas and to develop the area as a recreational and artistic milieu. In the first phase, the Development Programme began renting spaces for various purposes through the *Väliaikainen Hiedanranta* (Temporary Hiedanranta) operating model (Figure 2). The implementation was carried out by a newly established service design company that sought actors to raise local awareness of Hiedanranta by organizing events, attracting visitors, and increasing media visibility. The aim was to offer "a place for citizens, businesses and communities to develop a vibrant, versatile and sustainable urban culture" that would also support the future development of the area [24]. Among the early actors was the skateboarders' association Pirkanmaan Kaarikoirat—the 'Ramp Dogs' identifying with the broader region of Tampere—whose background and actions are introduced next.

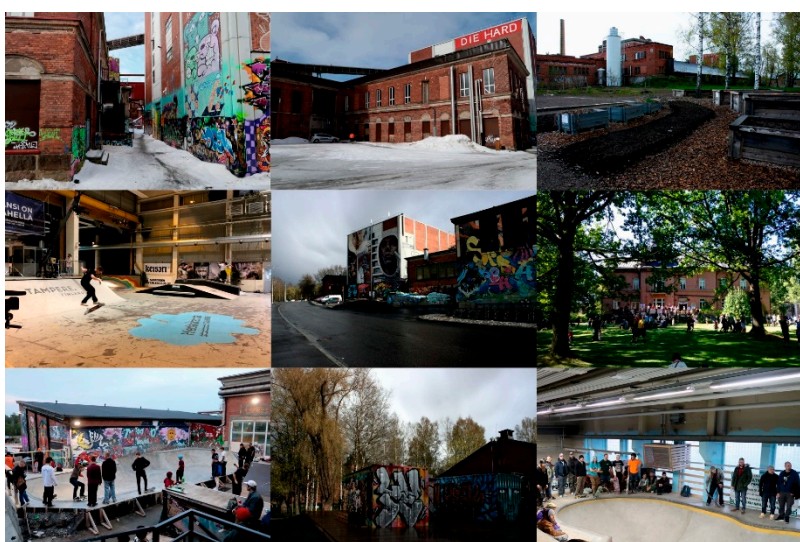

**Figure 2.** 'Temporary Hiedanranta' operating model has opened the Hiedanranta district for new activities, including the development of new skating spaces and events, street art exhibitions, culture events, and urban agriculture projects carried out by local citizens, communities, and businesses (photos by authors).

### 2.2. Skaters' Proactivity Enhancing Multistakeholder Governance

During the writing of this paper, the city of Tampere applied for the European Capital of Culture 2026 status, unsuccessfully yet with long-term development scenarios in mind. The Hiedanranta district was one of the key features in the bid, with 'Skate-Friendly City'

among the lead projects [25]. The theme was not an ad hoc idea as an active skate scene has been present in Tampere since the mid-1980s.

Skateboarding started to gain popularity in the urban life of Tampere some forty years ago. While some ramps were built by the city in the late 1980s, until the late 1990s skateboarding fell between different administrative sectors within the city organization, leading to lack of responsibility for its development. In response, the skaters' association Tampereen Lavetti was formed in 1990, to promote skateboarding in collaboration with the city, which led to the establishment of a new administrative model by end of the 1990s. Tampereen Lavetti received support from the Culture and Leisure Services to construct and maintain an indoor skatepark, while the Parks Department took charge of developing outdoor skateparks. Through this administrative model, which remains in operation today, seven skateparks including one indoor park were built by the early 2000s.

An increasing interest in skateboarding led to further demands. The draft of an ambitious plan with 13 new skateparks in the area was published in 2008. The participatory planning process involved numerous city departments, as well as schools, the local youth parliament, and the Lavetti association. However, the plan was never implemented. The disappointment related to this formal process, and a global spread of Do It Yourself (DIY) enthusiasm in skate culture, led a new generation of skaters to pursue novel ways to influence the local skate scene in Tampere. In 2012, a new skaters' association, Pirkanmaan Kaarikoirat, was formed, which collaborated with the city to plan a new skatepark—*Iso-Vilunen*—on an old gravel extraction site some six kilometers from the Tampere City Center (see Figure 1). The association contributed significantly to its design [26] and when the park opened in the spring of 2015, with a street course and a versatile bowl section, it was the first fully concrete skatepark in the Tampere region.

After that, Pirkanmaan Kaarikoirat has extended the urban collaborative network to include not just Culture and Leisure Services and the Parks Department, but also the city's employment services, sports and youth services, and several urban development projects—ranging from school yard renovations to Hiedanranta. Moreover, the collaboration is not limited to the city; many companies, associations, organizations (e.g., Youth Against Drugs), and faith-based institutions (e.g., Federation of Evangelical Lutheran Parishes in Tampere) are involved in the skateboarding initiatives. The connection to upper secondary education, analyzed in this paper, is the latest pioneering venture.

Pirkanmaan Kaarikoirat started to become involved in the Temporary Hiedanranta project in 2016. As they discovered an opportunity to build skateboarding facilities and organize events in an old factory building, they proposed a plan to the city that, in turn, saw skateboarding as a way to reach their set goals. The city's open-minded approach to developing Hiedanranta "together with residents, businesses and communities" [19] provided an opportunity to discuss the development of a larger skatepark facility containing indoor and outdoor sections and an event hub. The plan involved the transformation of an old industrial hall into what came to be known as the Kenneli DIY indoor skatepark [27]. This was a bigger project than anything before. The skaters had gained the adequate skills to design skateparks and the capability to gather the resources needed in the construction work. At present, the skatepark consists of indoor and outdoor concrete bowl-type skateparks, a vert ramp, a street course, and a new indoor street park to be opened in the fall of 2021.

### 2.3. Joint Learning Opportunities in Hiedanranta

The Sampo Upper Secondary School (SUSS), specializing in sports and communication, will open a new study track in the fall of 2021 focused on skateboarding and urban culture, including multimedia communication and event production. At the same time, the Hiedanranta Development Programme and Pirkanmaan Kaarikoirat are extending their collaboration to involve the Bachelor Program in Sustainable Urban Development (SUD) at Tampere University. The transdisciplinary SUD program—including tracks in Administrative Sciences, Social Sciences, and Technology—began in 2020, as the university's

response to the need for education for sustainable development (ESD). In parallel, a chair of Environmental Pedagogy was established at the university, contributing towards the same goals.

The development of the SUSS skateboarding program started in 2017. Inspired by the Bryggeriet High School in Malmö, Sweden [28], and recognizing the opportunities offered by the year-round skating facilities of Hiedanranta, the skaters involved with Pirkanmaan Kaarikoirat contacted the Director of Upper Secondary Education in Tampere who linked them with SUSS. The endeavor was encouraged by the association's previous experience of combining skateboarding and education with the Tampere Vocational College. In November 2018, two city councilors proposed an initiative of the skateboarding program in upper secondary education and vocational training [29]. It subsequently passed through the decision-making processes of the city board and was accepted by the committee for competence and economic development responsible for general upper secondary education. In November 2020, the city council approved the budget for 2021, which confirmed the launch of Finland's first skateboarding-oriented study program in August 2021 [30]. Students from all around Finland can apply to the program and the school is easily achieved at the Tampere city-region.

The formal decision-making process and the planning of the program curriculum was conducted in close collaboration with Pirkanmaan Kaarikoirat and the wider skate community. Partnerships with skate associations in other Finnish cities, the Finnish Skateboarding Association, the Finnish Female Skateboarders Association, and the Finnish Olympic Committee, were established along the way. Concurrently, connections with several private sector actors as well as international skate organizations were formed. One specific outcome is that some of the students may choose to focus on competitive skateboarding with the aim of joining the national Olympic team. To support this, the Finnish Olympic Committee, the Varala Sports Academy, and the Finnish Skateboarding Association have started to train a group of skateboarding instructors to work in the skateboarding program [31]. More broadly, the partnerships offer students opportunities to integrate with different sectors of the society in the local urban context as well as to join in skate communities nationally and internationally. Indeed, one of the main aims of the program is the enhancement of 'skateurbanism', where connections have been made with the cities of Malmö and Stockholm in Sweden, Bordeaux in France, and Nottingham in England—including further collaboration opportunities with SUD. Overall, the skateboarding high school initiative is in line with the national curricular framework emphasizing increasing connections, not only to higher education, but also to businesses and working life [32].

In the skateboarding high school initiative, joint learning opportunities are being developed for SUSS and SUD students. They make use of the existing networks of collaborators involved in the development of Hiedanranta. These learning opportunities will be organized around physical and social conditions related to skateboarding, including the versatile multi-purpose historical industrial space; physical skateboarding objects; embodied experience space; skaters' local, national, and global networks; political economy of a new district where skating is seen as positive activity; climate mitigation and restoration of biodiversity in a built environment; urban physical processes; actor network and interest groups tied to skating; aesthetic and visual dimension; as well as stories, discourses, and representations related to skateboarding. The systematic connections between these elements form what we call the *Hiedanranta skatescape*. As we will discuss in Section 4, it entails the essential properties of a "boundary object," [15,33] being versatile and constantly shifting but also identifiable and concrete enough to function as a node for different framings and practices, partly based on different rationalities yet driving towards the same direction.

In practice, the joint learning opportunities will include learning events, placemaking, and co-creation practices around a new skateboarding area, where high school and university students can learn about and participate in the efficient use of resources, enhancement of local biodiversity, sustainable community building, and documentation of changes in

their immediate environment by audiovisual means. The collaboration is driven by a recent national reform of the *Act on General Upper Secondary Education* [32,34], which aims to facilitate a smooth transition from upper secondary education to higher education. It also aims to meet the ambitious goals of the renewed high school curriculum that promotes the sustainable transformation of society.

### 3. Co-Creating Sustainable Urbanity through the Hiedanranta Skatescape

#### 3.1. Co-Creational Research Approach

The transdisciplinary development described in the paper is based on the authors' long-term acquaintance with co-creational research, particularly in urban studies contexts, including collaboration with various professionals, policy makers, private actors, NGOs, grassroots actors, and other citizens involved in city-regional planning and development [5,35–39]. Polk [40] describes aptly the basic idea of transdisciplinary co-production of knowledge where real-life problem solving is targeted concurrently with scientific aims:

"Knowledge is co-produced through the combination of scientific perspectives with other types of relevant perspectives and experience from real world practice including policy-making, administration, business and community life. Co-production occurs through practitioners and researchers participating in the entire knowledge production process including joint problem formulation, knowledge generation, application in both scientific and real-world contexts, and mutual quality control of scientific rigor, social robustness and effectiveness."

Additionally, the first author has long-term experience of co-creational research with professionals working with children, youth, and families, including youth workers, teachers, care workers, social workers, welfare and health professionals, and administrative personnel from all these fields [41–43]. In this line of research, she has developed a theoretically informed practical approach of Positive Recognition, in multi-disciplinary and multi-professional collaboration [44]. The co-creative work seeks to advance wellbeing among children and young people as well as to prevent marginalization in the society. The approach emphasizes the active role of citizens, or 'ordinary people', in research, on a par with the professionals working with them. As Satka et al. [45] state, in co-creative research "each actor occupies the position of a knowing subject".

In her recent applied research, the first author has brought the principles of Positive Recognition to inform her co-creative research with urban planners and policy makers, to develop means for citizen participation in city-regional development processes [20,37,46]). Drawing from this long-standing methodological inquiry, she is currently bringing the cumulative understanding into environmental pedagogy with three specific foci: positive recognition in the pedagogies of climate change, biodiversity, and sustainability. In what follows we will describe how Positive Recognition can enhance sustainability education in the university and high school levels, through the Hiedanranta skatescape that we are developing as a pedagogical boundary object, involving importantly university and upper secondary school students as "knowing subjects" along with other citizens, professionals, business actors and city representatives, as co-producers of knowledge. The paper hence notably focuses on responding to transdisciplinary methodological objectives.

#### 3.2. Positively Recognizing Sustainable Urban Pedagogy

Positive Recognition draws from the ethics and politics of recognition theorization, with Charles Taylor [47], Axel Honneth [48], and Nancy Fraser [49] as influential philosophers. In the spirit of Hannah Arendt's [50,51] political philosophy, 'recognition' refers to socially embedded constitutive relations between individuals and groups that, when successfully accomplished, contribute to creating and maintaining a society characterized by *experienced equitability* and *acknowledged diversity*. Simply put, by recognizing each other correctly, people can make life more ethical through shared experiences of belonging, and, vice versa, misrecognition leads to confrontations and biased categorizations that divide people in their mundane communities and in society at large.

In practice successful recognition may mean, for example, that persons from different generational positions can share a public space in mutual understanding even if their use of the space differs notably, e.g., children playing, youth skating, adults having business lunches, and elderly persons resting their feet at a square where everyone can feel safe and welcome. In urban life, such sharing does not always come about organically but requires supportive measures, and similarly, consensus in urban development projects may be hard to reach. The educational collaboration project between SUD and SUSS takes this challenge as its primary focus by engaging different actors involved in the Hiedanranta district development in a positively recognizing spirit, and, thus, co-creating practical knowledge on sustainable urban development with the university and the upper secondary school students.

As an operational principle, Positive Recognition sets out to strengthen dignity and inclusion in people's day to day environments, following the ideals of social sustainability. It offers thinking tools for understanding, exposing, and engaging with everyday communal dynamics, and tackling divisive issues. As a democratic ideal, the recognition perspective makes visible societal structures and power relations that cannot be erased but can be contested and changed; by emphasizing *experienced* equitability and *acknowledged* diversity, it encourages a practice-oriented critical stance. Drawing particularly from Honneth's [48] theorization, Positive Recognition denotes that reciprocal *care* in personal relationships, participatory *inclusion* in everyday communities, and *respect* for equality and difference in the society are imperative to the development of people's self-confidence, self-esteem, and self-respect. These, in turn, provide for active agency and empowerment. In our case, the sustainable pedagogy emphasizing these basic human needs—for care, acknowledgement, and respect *over* difference *as* equals—supports urban agency that may unfold in different contexts of private and public life.

Positive Recognition draws attention to the contextuality and situatedness of sustainable agency and, in this sense, takes forward the joint traditions of critical pedagogy and critical geography [52–55]. It is a dynamic driver of democracy, resonating with major issues of social and political life, such as gender [56], race [57], class [58], and sexuality [59], but also environmental relations that remain largely missing in the existing research inspired by recognition theories. We consider recognition a productive concept for rethinking sustainability pedagogies particularly because it reaches beyond simple left-right and liberal-conservative positionings on the political map, allowing an issue-based approach where the environment and the other three key dimensions of sustainability—namely economic, social, and cultural—are considered as the core values of the sustainable city. It helps in "bringing issues back to participation", as Leino and Laine [35] suggest, and supports the development of issue-based citizenship, as Häkli et al. [46] proposes.

Recognition-informed sustainability education acknowledges giving and receiving respect as a starting point of democratic urban development, among city-dwellers and within multistakeholder governance. Hence, all pedagogical endeavors should be based on, first, learning about different people's perspectives regarding the issue at hand; second, acknowledging people's contextual and situated ways of knowing as valuable per se, especially when it comes to issues of particular importance to them; and third, supporting sustainable agency on the basis of positively recognized experiential knowledge that allow for diversity between individuals and groups. Through this pedagogical approach, trust between different actors can be established and strengthened, which is a key element in the pedagogical use of the 'boundary object'.

### 3.3. Hiedanranta Skatescape as a Boundary Object

What we term the Hiedanranta skatescape is a boundary object that facilitates encounters between different actors in the spirit of positive recognition. According to the famous definition by Star and Griesemer [33], "boundary objects are objects which are both plastic enough to adapt to local needs and the constraints of the several parties employing them, yet robust enough to maintain a common identity across sites". This notion was first introduced as part of a study that focused on the efforts of Joseph Grinnell, the first director of the

Berkeley's Museum of Vertebrate Zoology, to establish the museum together with people from different social worlds, including university administrators, curators, amateur collectors, and researchers. While these groups had different agendas, the museum functioned as a flexible yet identifiable object around which the labor of all could be organized.

The concept of the boundary object has been applied in many fields of research, including urban planning, education, and sustainability science, which involve complex problems, multiple stakeholders with partly conflicting goals, and the transmission of information and knowledge across disciplines and communities [15,16,60–62]. Boundary objects can be physical objects or repositories; "good enough" maps or diagrams without detailed information; or methods used consistently by different groups [33]. A specific object can be mobilized for diverse purposes and may have different meanings in different social worlds or communities of practice. Yet, identified by various actors as *the* object, it serves as a starting point and meeting ground for debate and cooperation.

In the multi-stakeholder governance and transdisciplinary education of sustainable urban development, boundary objects can be used for enhancing mutual appreciation across diversity, with the aim of challenging dominant discourses and power structures. As Akkerman and Bakker [15] state, in "boundary literature [ . . . ] the emphasis is on overcoming discontinuities in actions or interactions that can emerge from sociocultural difference rather than overcoming or avoiding the difference itself." For instance, the governance of urban planning may be challenging if all parties attach different meaning to a planning project and have distinct ways of participating. Kanninen [60] demonstrates this with the case of the planning of a new tramline in Tampere, and how its technical implementation plans have functioned as boundary objects that have facilitated coordination and collaboration. Häkli [61] analyzed cross-border collaboration in the establishment of the transnational twin-city of Haparanda-Tornio by identifying the border river as a boundary object of urban planning and governance, beneficial for the collaborative practices but also more generally for trust-building. In the same way, the development process of the Hiedanranta skatescape brings together actors from different quarters, in ways that enable the gradual development of trust.

As the ongoing collaboration in Hiedanranta demonstrates, the skatescape has proven to be a useful boundary object for individual skaters and their DIY-based association, various departments of the City of Tampere, residents and other local actors, construction companies involved in the development process, and the study programs of SUSS and SUD. These actors approach the skatescape from different perspectives with partly tangential interests. First, for skaters, the skatescape has provided an opportunity to develop a local skill base and to scale up their work. Due to the perseverance and versatility of the local skate community, skateboarding has gained a strong foothold in Tampere, resulting in an active international community via social media and national and international publicity [27,63,64]. Although skaters have partnered with the city for over 30 years—in urban planning, participatory budgeting, and urban development—the DIY ethos remains strong among them. The Hiedanranta skatescape fosters this versatility; to work with the city proactively *and* to build and use things in a self-organized DIY fashion. The fact that these formal and informal activities are not mutually exclusive is an achivement in its own right; bringing together Habermasian communicative action and Rancièrian dissensus is never an easy task but requires constant negotiation [36,65].

Second, for the City of Tampere and the Development Programme of Hiedanranta, the skatescape has embodied the 'maker culture' of the skater community and other related grassroot organizations, such as the electronic music and culture association *SWÄG*, circus association *Faktori*, *Tampere Photography Association*, art galleries *Hieta* and *Muuntotila*, the street art association *Spraycankontrol,* and the artisanal and artist collective *Paja*. The city has embraced the possibility to transform the old industrial site into a culturally unique, multi-purpose, historical area. In this context, the skatescape contributes to enhancing local, national, and international awareness of a thriving and edgy, socially sustainable district. Skateboarding events and competitions like *Manserama* and the *Finnish Skateboarding*

*Championships* involve skaters from all around the world and receive publicity, especially through social media outlets with a broad scope, but also through the traditional media (e.g., the international newspaper *Guardian*, Finnish broadcasting company *YLE*, regional newspaper *Aamulehti*).

Third, for profit-seeking construction companies, the skatescape is part of a unique architectural and cultural-historical milieu with a distinct reputation and promising branding capacities. As international examples show, a strong brand is generally perceived to enhance the desirability of an area and, thereby, to increase the stability of development projects through risk reduction and profit maximization [66] (reference removed for the review process). In the case of Hiedanranta, the leisure amenities and flexible spaces are seen to add value to the district by attracting "active and entrepreneurial people" [67]. While concrete collaboration with them is yet to begin, many actors in the business identify the potential embedded in the Hiedanranta skatescape.

Fourth, the positive ambience and publicity of the skatescape have offered synergy gains, as well as learning and networking opportunities, to a large group of other actors. These include: *Visit Tampere* (tourism, event and congress organization); schools and youth services; the Tampere Vocational College Tredu; culture and leisure services and professional event producers; local companies ranging from media companies and breweries to circular economy start-ups; and a wide range of actors in the Finnish and international skateboarding scene, including the above mentioned grassroots organizations, several board and clothing companies, skate shops, skate media, film makers, skatepark construction professionals, and skate organizations from Finland and beyond. Figure 3 portrays some examples of what the Hiedanranta skatescape offers: learning possibilities for children, a platform for international skateboarding events, and an urban development project that has brought together professional builders and DIY-enthusiasts while attracting also international visitors and gaining media attention.

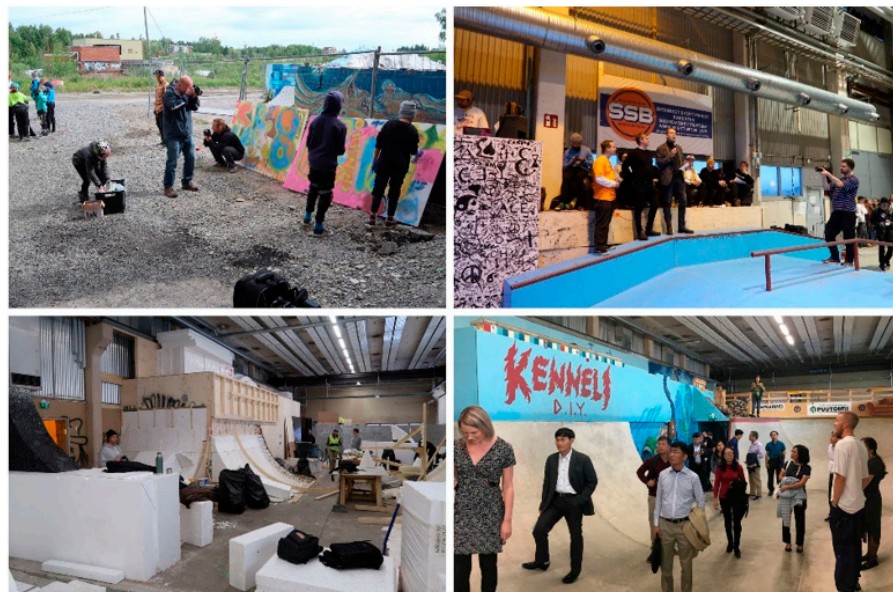

**Figure 3.** Various synergies in Hiedanranta: At the left bottom corner, professional skatepark construction companies and the Finnish Skateboarding Association are involved in the building of the Kenneli DIY; above that, the national broadcasting company is making a news piece of a children's skate school. At the right-top corner, the city's director of Education and Culture, Lauri Savisaari, is giving a speech during the Finnish Skateboarding Championships, concurrently documented by the HangUp skatemedia. Below that, a group of Vietnamese government and administrative officials are learning about the Kenneli DIY, as part of their visit with Tampere University. (Photos by authors).

Finally, for the educational institutions examined in this paper, the skatescape involves multiple opportunities in terms of curriculum content and collaboration. In keeping with

the national upper secondary education requirements, the SUSS skateboarding program uses the skatescape to advance specific studies on different elements of skate culture, such as videomaking, photography, event production, and skatepark design and construction. Not least importantly, such a program invites students who otherwise might not enter the academic study path. The broader themes of urban culture, social agency, and sustainability inform the study content. The overarching aim is to support students' capabilities to act in society, which links it strongly with democracy education. Through collaboration with SUD, the SUSS skateboarding program introduces university-level studies to high school students, which aims at lowering the threshold for these youths to continue studies in higher education. The collaboration is a concrete example of the aims of the renewed Act on General Upper Secondary Education—to ease the transition to higher education and to increase the share of university graduates among 25–34-year-olds from 41 to 50 percent [14].

For SUD, the Hiedanranta skatescape provides hands-on experience on grassroots movements and DIY culture tied to urban development and collaborative planning, along with the broader urban collaboration. As we will next show, the joint learning opportunities around the skatescape contribute to achieving learning outcomes related to, first, collaborative work, second, responsibility in relation to sustainable urban development, and third, accessible communication among peers, teachers, and external audiences, across disciplines and professions.

### 3.4. Connecting Students and Stakeholders through the Urban Lab

In the transdisciplinary and cross-institutional urban education project, SUD and SUSS will build on and expand the existing plurality of encounters and perspectives in Hiedanranta by using the skatescape as a recognized common ground for students and stakeholders—that is, by mobilizing it as a *positively recognizing pedagogical boundary object* through the Urban Lab course in the SUD degree program. The course provides students with a transdisciplinary learning experience by enabling them to undertake real-life projects related to urban planning, community work, designing, and crafting of objects for public use, following the co-creational research approach. The approach encourages proactive engagement with stakeholders, in this case including—importantly—the multi-scalar skate community, along with the public, private and third sector actors involved in the Hiedanranta Development Programme.

The Urban Lab course begins with a research phase where the viewpoints, knowledge bases, and skills of the students related to the Hiedanranta skatescape are explored and duly recognized. This includes identifying critical issues regarding the 'politics of living together' and community building in the Hiedanranta district, which makes space for youth-initiated democratic urban development. The acknowledged student perspectives form the starting point to the next phase where we identify significant citizen groups involved or affected by the raised issues. In the spirit of recognizing positively people's contextual and situated ways of knowing, we gather data with different methods: desktop study of other cases in Tampere and beyond; observing urban hitchhiking [68] interviews; workshops; and exploring embodied ways of experiencing space. Our ambition is to build a versatile knowledge base for creative urban development, following the ideals of positive recognition.

As an example, our previous research related to citizen engagement in city-regional development is currently being used in a 'regional citizen panel' project led by the City of Tampere, where 'regional traffic and mobility' is considered [46]. The idea is to bring city-regional dwellers from different social positions and geographical locations into a shared virtual space where they can raise critical issues, related to mobility in the region and to using different forms of transport, including importantly sustainable light traffic and public transportation. The work of the panel is facilitated by professionals who understand how this knowledge can be used in related urban planning and administration processes, and who direct discussion towards potentially impactful results. Rather than controversial debates about different forms of transportation, this citizen-professional co-working seeks to recognize people's variable concerns and find solutions to address them in the ongoing

planning and administrative processes, in the spirit of positively recognizing co-creational knowledge production.

The Urban Lab has similar, but more local objectives: to identify relevant issues in the Hiedanranta skatescape with the students; to hear a multiplicity of voices related to them across the public-private-civic sphere, concurrently building trust amongst the participants; to facilitate communication between these ideas and the involved actors through the boundary object of the skatescape; and to come up with solutions that make the Hiedanranta district a better urban space from multiple perspectives. To summarize, the Urban Lab course seeks to meet the following three aims:

(1) To create *opportunities for co-learning* and to the related development of *respectful project management skills*, where university and high school students have different roles. By creating shared visions around the tangible character of the skatescape, they will learn to contribute to a common development process and manage multiple perspectives, rather than selecting one perspective that either explicitly or implicitly dominates the others.

(2) To strengthen the *socio-environmental dimensions of the Hiedanranta skatescape*. In practice this may mean that, while recycling the skate elements and materials and improving the skate site, we will concurrently make improvements to the natural environment. Additionally, the social dimensions of sustainability are strongly emphasized, through the perspectives of equity, diversity, inclusion, accessibility, and social cohesion. Again, bringing together multiple perspectives and local needs enables the parties involved to come up with novel ideas of what constitutes a good democratic urban environment, which may also create new opportunities for livelihood.

(3) To use the skatescape as a pedagogical boundary object to facilitate the *mobilizaton of critical urban development theory*. Lefebvre's idea of the "right to the city", for instance—entailing that anyone has the right to participate in the making of the city regardless of property rights or ownership—can be reframed in ways meaningful to the parties involved through the shared element of the skatescape [69]. From another theoretical pespective, the collaborative work can be approached as Arendtian politics of living together [50,51], characterized by reciprocal care, participatory inclusion, and respect for equality and difference a là Honneth [48]. SUD students can use these theoretical ideas as lenses to explore alternative ways of approaching urban development, participation, and engagement of citizens. For SUSS students, these ideas can lend new insight into their relationship with skateboarding, urban space, and the right to act in ways meaningful to them, in respect of other urban dwellers.

The Urban Lab course hence offers opportunities for dialogue between the two study groups, local and regional stakeholders, the city, the broad skate community, and citizens in the Tampere city-region. The SUD students will learn to communicate sustainability issues to different audiences and to relate productively to various issues and actors, including social (e.g., gender in public space), environmental (e.g., development of land use, zero carbon solutions, biodiversity issues, and traffic), economic (e.g., for-profit private actors in urban development, sustainable start-ups), and cultural (e.g., skateboarding as an urban practice) sustainability dimensions. In parallel, the SUSS students will learn to verbalize and reflect upon the aspects of their sub-culture with relevance to sustainable urban development, in connection with the stakeholders and the scholarly community. Moreover, in the spirit of co-creational research, this dialogue and exchange of ideas enables teachers, students, and stakeholders to explore theoretical ideas on sustainable urban development at different levels of conceptual complexity.

## 4. Conclusions

In this paper we have shown, through a case study, how urban development, DIY skate culture, and higher and secondary education can be linked in broad stakeholder collaboration towards a sustainable city. The introduced co-creative, transdisciplinary educational project includes two institutions and study programs from different educational levels, several divisions of the city, a number of informal actors, private companies from different sectors

of economic life, and the civil society ranging from local to global social communities. As a key element enabling the productive collaboration, we have identified the *skatescape as a pedagogical boundary object* that invites different actors to recognize each other in a positive manner in the Hiedanranta context. This methodological finding offers insight, not only to sustainability education, but also more broadly to positively recognizing environmental pedagogy, contributing to critical pedagogy and critical geography scholarship.

Drawing from previous research on the ethics and politics of recognition, we have introduced a novel pedagogical perspective to sustainable urban development, developed in connection with the transdisciplinary and cross-institutional Urban Lab project. The paper proposes *positively recognizing sustainable pedagogy* to critical urban education, as part of a broader framework of positively recognizing environmental pedagogy including climate and biodiversity education. With this approach, the importance of binding urban development together with democratic urbanism is highlighted, which links firmly the dimensions of environmental and social sustainability, as well as the economic and cultural aspects that have a notable role in the Hiedanranta skatescape.

One important actor in the establishment of the joint study programs is a local, nowadays globally acknowledged skaters' association, Pirkanmaan Kaarikoirat. Over the past 30 years, the activities of the local skaters have included broad-ranging collaboration with public and private actors as well as DIY activism that disengages from such relations. This combination of communicative and agonistic urban action—a simultaneous search for collaboration and acceptance of confrontations as integral to pluralist societies—sustains transformative power among the skate community and contributes to trust-building with various quarters. The local skaters may take part in both or either types of activities, which upholds plurality and open-mindedness among them.

In our pedagogical approach, we lean on and take further this non-essentialist approach to urban political agency. The skater community and its collaborators have shown how urbanity can be developed co-creatively without fixed aims or a shared ideological basis. These endeavors go in line with a city where people with different age, cultural background, socio-economic position, and preferences can live together through *experienced* equitability and *acknowledged* diversity. The transdisciplinary, cross-institutional education project introduced in this paper seeks to contribute to such progress from the perspectives of sustainable pedagogy and multistakeholder governance.

**Author Contributions:** Conceptualization, K.P.K.; writing—original draft preparation, K.P.K., M.K., S.J., M.L. and J.T.; writing—review and editing, K.P.K., M.K., S.J., M.L. and J.T. All authors have read and agreed to the published version of the manuscript.

**Funding:** This research received no external funding. The APC was funded by four Tampere University faculties: Faculty of Education and Culture, Faculty of Social Sciences, Faculty of Management and Business, and Faculty of Built Environment.

**Institutional Review Board Statement:** Not Applicable.

**Informed Consent Statement:** Not Applicable.

**Data Availability Statement:** Not Applicable.

**Acknowledgments:** We are indebted to our interdisciplinary research environments POLEIS and RELATE that have brought us together over the years; thanks to everyone who has taken part in the research seminars where lively discussions on the key topics of the paper have taken place, this truly is an outcome of our joint efforts. We also stand thankful for our practical collaborators who have offered their insight to our co-creative work in various research and development projects. Finally, we wish to thank the editors and the reviewers of the journal for their encouraging attitude on our paper, and our Faculties for supporting this OA publication.

**Conflicts of Interest:** The authors declare no conflict of interest. The funders had no role in the design of the study; in the collection, analyses, or interpretation of data; in the writing of the manuscript, or in the decision to publish the results.

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
