# Peer review of "Skatescape in the Making: Developing Sustainable Urban Pedagogies through Transdisciplinary Education"

_sustainability, doi:10.3390/su13179561_

Round 1

Reviewer 1 Report

overall, v.good article, interesting and easy to follow

1- please assign a section to the research questions and research method

2- proofreading and grammar check before the final submission.

Author Response

In the introduction, we have specified the set aims of our joint endeavor in the form of a research question.

We have added a sub-section that describes our methodological approach, titled "Co-creational research approach".

One of the authors is a native speaker of Canadian English and he has gone through the paper once more for idiomatic US expression.

Reviewer 2 Report

This is a very well written paper on a highly relevant topic, with a well structured and supported argument.

The paper's main strength and innovation lies in combining insights from pedagogy with urban studies, political philosophy, museology, and environmental science. Ideally, the study would include empirical data from the project implementation stage. As the implementation is yet to happen, at the moment what the paper provides is primarily the theoretical framework and literature review. However, it is capable of standing on its own due to the strength of its main argument and is therefore recommended for publication.     

Author Response

No requests for revisions have been presented by the reviewer.